

# Measuring prairie snow water equivalent with combined UAV-borne gamma spectrometry and lidar

Phillip Harder[1], Warren D. Helgason[1,2], John W. Pomeroy[1]

[1]Centre for Hydrology, University of Saskatchewan, Saskatoon, Saskatchewan, Canada
[2]Department of Civil, Geological, and Environmental Engineering, University of Saskatchewan, Saskatoon, Saskatchewan, Canada

*Correspondence to:* Phillip Harder (phillip.harder@usask.ca)

**Abstract.** Despite decades of effort, there remains an inability to measure snow water equivalent ($SWE$) at high spatial resolutions using remote sensing. Passive gamma ray spectrometry is one of the only well-established methods

to reliably remotely sense $SWE$, but airborne applications to date have been limited to observing km-scale areal averages over shallow snowcovers. Noting the increasing capabilities of unoccupied aerial vehicles (UAVs) and miniaturization of passive gamma ray spectrometers, this study tested the ability of a UAV-borne gamma spectrometer and concomitant UAV-borne lidar to quantify the spatial variability of $SWE$ at high spatial resolutions. Gamma and lidar observations from a UAV were collected over two seasons from shallow, wind-blown, prairie snowpacks in

Saskatchewan, Canada with validation data collected from manual snow depth and density observations. The ability of UAV-gamma to resolve the areal average and spatial variability of $SWE$ was promising with appropriate flight characteristics. Survey flights flown at a velocity of 5 m s[-1], altitude of 15 m, and line spacing of 15 m were unable to capture the average or spatial variability of $SWE$ within the uncertainty of the reference dataset. Slower, lower, and denser flight lines at a velocity of 4 m s[-1], altitude of 8 m, and line spacing of 8 m were able to successfully observe

areal average $SWE$ and its variability at spatial resolutions greater than 22.5 m. Using a combination of UAV-based gamma $SWE$ and UAV-based lidar snow depth improved the results substantially and permitted estimation of $SWE$ at a spatial resolution of greater than 0.25 m with a ±14.3 mm SWE error relative to manual snow survey density and UAV-lidar based depths to estimate $SWE$. UAV-borne gamma spectrometry to estimate $SWE$ is a promising and novel technique that has the potential to improve the measurement of shallow prairie snowpacks, and when combined

with UAV-borne lidar snow depths, can provide high resolution, high accuracy estimates of prairie SWE. Research on optimal hardware, data processing, and interpolation techniques is called for to further improve this remote sensing product and explore its application in other environments.

## 1 Introduction

Snow is a defining feature of the hydrological cycle in cold regions and has significant socioeconomic and
environmental implications (King et al., 2008; Pomeroy and Goodison, 1997). A basic and persistent challenge for snow hydrology is efficiently and accurately quantifying snow water equivalent. The overlapping variability of landscape, weather and climate, and snow processes combine to drive significant spatiotemporal differences in snowpack characteristics (Essery and Pomeroy, 2004b; Grünewald et al., 2010; Liston and Sturm, 1998; Pomeroy and



Gray, 1995; Shook and Gray, 1996; Trujillo et al., 2007) A significant body of research has been devoted to developing

protocols and technologies to observe snow characteristics to inform scientific understandings and decision making
(Kinar and Pomeroy, 2015). Quantifying the spatial variance of snow water equivalent ($SWE$) allows calculation of
snow covered area (SCA) depletion during the melt period (DeBeer and Pomeroy, 2010; Essery and Pomeroy, 2004a;
Faria et al., 2000) and in turn the SCA depletion is critical to estimate the contributing area, and duration, of runoff
and infiltration from snowmelt (DeBeer and Pomeroy, 2010; Shook et al., 1993). To date the ability to directly and

remotely observe the spatial variability of $SWE$ at the fine scales corresponding to the snow redistribution and ablation
processes defining snowpack formation has remained elusive (Tedesco et al., 2015).

The $SWE$ (water equivalent water depth per unit area) of a snowpack is expressed in mm water equivalent or kg
m$^{-2}$. Snow surveys of depth ($h_s$) and density ($\rho_{snow}$) observations along a linear transect are the traditional approach
used to calculate $SWE$, and remain the most reliable technique, but are time consuming, labour intensive, and

ultimately a destructive sampling technique (Kinar and Pomeroy, 2015). Non-contact point scale observations such as
snow pillows, passive radiometric sensors, and acoustic sensors have demonstrated success but do not capture spatial
variability (Coles et al., 1985; Kinar and Pomeroy, 2007, 2015; Wright et al., 2011). Remote sensing has had great
success in quantifying the spatial variability of $h_s$ over wide ranges in extent and resolution ranging from satellite
stereography (Marti et al., 2016), lidar (airplane-borne or UAV-based) (Deems et al., 2013; Harder et al., 2020;

Hopkinson and Collins, 2009; Jacobs et al., 2021), and structure from motion techniques (Bühler et al., 2016; Harder
et al., 2016; Walker et al., 2021). Snow depth observations alone capture a significant amount of the snowpack
variability but need additional observations, or estimation, of $\rho_{snow}$ in order to quantify $SWE$ (Painter et al., 2016).
$SWE$ remote sensing products tend to be coarse scale, utilizing passive microwave or gamma remote sensing (Tedesco
et al., 2015; Tong et al., 2010; Tuttle et al., 2018), or active radar sensors (Tsang et al., 2021) .

Gamma remote sensing of $SWE$ relies on two principles. First, all soils contain naturally occurring gamma particle
emitting radioisotope elements (Topp, 1970). Second, mass, including all phases of water, attenuates gamma radiation
(Peck et al., 1971). Beer's Law, which relates the transmission of radiation through a medium ($I$) to the intensity of
the source ($I_0$) as an exponential function of the attenuation coefficient ($\mu$) and thickness ($d$) of the attenuating
medium, as

$$I = I_0 e^{\mu d},\tag{1}$$

can be adapted to estimate $SWE$ from observations of gamma emissions over time. By using count rates of gamma
particles above a surface when snow-covered ($C_{snow}$) and snow-free ($C_{bare}$) in place of $I$ and $I_0$, respectively, and
assuming a $\mu$ for water (5.835 10$^{-3}$ mm$^{-1}$, Carroll (2001)), the $d$ can be interpreted, and solved for, as $SWE$ (mm) as,

$$SWE = -\frac{1}{\mu}\ln\left(\frac{C_{bare}}{C_{snow}}\right).\tag{2}$$

This requires an assumption of isotropic gamma emissions from the soil and no change in soil water content in the
time between the bare and snow-covered surface observations that would change $C_{bare}$ (Carroll and Carroll, 1989).
Two main limitations are inherent in quantifying $SWE$ with gamma approaches. The first is that high attenuation of
gamma rays by water leads to complete attenuation of the gamma signal in large snowpacks, such that this technique
is limited to medium or shallow snowpacks. In a point scale/stationary implementation the Campbell Scientific CS725

passive gamma radiation sensor (Kinar and Pomeroy, 2015; Wright et al., 2011) when fixed above a snowpack can



estimate $SWE$ for footprints of 50-100 m$^2$ at 3 m sensor height with 15% accuracy and is limited to snowpack's with <600 mm $SWE$. The CS725 has been shown to work well for uniform and relatively deep mountain snowpacks, if placed on mild slopes where snowmelt runs off instead of ponding (Smith et al., 2017). In an airborne implementation, the NOAA Airborne Snow Survey program has utilized gamma spectroscopy to observe peak $SWE$ over much the

Red River Basin of the north-central US Great Plains and southern Canadian Prairies to inform flood predictions since 1980 (Cho et al., 2019). This airborne program typically employs flight lines at 150 m altitude, 16 km long to provide $SWE$ estimates with approximately 5-7 km$^2$ footprints with errors less than 10% for snowpacks <300mm $SWE$ (Carroll and Carroll, 1989; Cho et al., 2019; Tuttle et al., 2018). The second limitation is that variability in soil moisture is a significant source of uncertainty. A snow-free observation to capture the background gamma state as near as

possible to freeze up is required. In the case of an overwinter increase in near surface soil moisture, due to snowmelt or rainfall infiltration, end of winter $SWE$ will be biased high (Carroll and Carroll, 1989). Approaches to correct for overwinter changes require independent estimates soil moisture change (Carroll and Carroll, 1989; Carroll, 2001; Offenbacher and Colbeck, 1991).

Passive radiometric observation methods are sensitive to an integration time and, in mobile applications,
challenged by small signal to noise ratios (Peck et al., 1971; Reinhardt and Herrmann, 2019). The ability to resolve a feature of interest with gamma spectroscopy is directly related to the volume of the scintillation crystal, integration time, and proximity to the target which all need to be balanced by the physical limitations and operational characteristics of the platform, area of interest, and ability to precisely collocate sensors between different surveys (Reinhardt and Herrmann, 2019). The confluence of ever-increasing UAV capabilities (endurance, payloads, and
spatial accuracy of navigation) and miniaturization of gamma ray spectrometers has opened the door to UAV-borne spectroscopy. Most UAV-gamma applications to date have focused on mapping radiative properties for mineral exploration (Martin et al., 2020) and relationships to soil properties such as texture, type, nutrient status, erosion, organic matter and pH (Reinhardt and Herrmann, 2019). A significant advantage of UAV platforms over traditional crewed aircraft is the ability to repeatedly fly consistent flight lines at low altitudes and speeds.

The ability of UAV-borne gamma spectroscopy to quantify $SWE$ has not been reported in the scientific literature, nor has the possibility to interface gamma-measured $SWE$ with high resolution snow depth observations from UAV-lidar been examined. The purpose of this work is to demonstrate the workflows needed for deploying UAV-borne gamma spectroscopy over snow and then to evaluate: 1) the ability of UAV-borne passive gamma spectroscopy to directly observe the $SWE$ of shallow agricultural snowpacks; and 2) the potential for UAV-borne gamma spectroscopy
by itself, and combined with UAV-lidar, to estimate the spatial variability of $SWE$ at fine spatial scales.

## 2 Data and Methods

### 2.1 Study Area

Observations were collected over two snow seasons between fall 2020 and spring 2022 southeast of Saskatoon, Saskatchewan, Canada, in an agricultural region of the Canadian Prairie ecozone. Two study sites were chosen with
both having low relief and hummocky topography (Table 1). The Stubble site is a cultivated field, seeded the previous





year with barley that was harvested in September leaving a 15 cm standing stubble. The perennial grassland, which is grazed during summer, contained grasses, fescues, shrubs, and forbs with a height ≤ 30 cm in fall 2021. As a result of drought conditions in summer/fall, field observations showed low near surface soil moisture contents at both sites and with dampened spatial variability in both years.

**Table 1. Summary of sites and observations**

| Site Name | Stubble | Grassland |
|---|---|---|
| Location | 51° 56.11' N | 51° 23.39' N |
| | 106° 21.99' W | 106° 26.12' W |
| Surface Condition | Standing barley stubble | Grass and small shrubs |
| | height 0.15 m. | height <0.3 m |
| Soil Texture | Loamy Sand | Sandy Loam |
| Snow Free Observation | Nov 7, 2020 | Nov 9, 2021 |
| Snow-Cover Observation 1* | Nov 13, 2020 (Fall) | Mar 14, 2022 |
| Snow Cover Observation 2* | Mar 9, 2021 (Spring) | |
| UAV flight profile characteristics | 5 m/s, 15-m altitude, 15-m flight line spacing | 4 m/s 8-m altitude, 8-m flight line spacing |

*bracketed identifiers denotes the specific observation for reference hereafter

## 2.2 Data Collection

### 2.2.1 Site Conditions and Surveys

Several UAV gamma surveys were made, concomitant with UAV-lidar surveys. Meteorological conditions during the respective seasons was observed using well-instrumented meteorological stations (part of the Global Water Futures Observatories www.gwfo.ca) near to the study locations. Each survey captured different environmental and deployment conditions. In fall 2020, a bare ground survey was conducted at the stubble site on November 6 immediately preceding 60 mm of snow (water equivalent) which fell over November 7-9. This provided an opportunity to test the $SWE$ estimation by conducting a subsequent snow-covered survey on November 13. For this survey interval there was a clear transition between exposed, unfrozen and relatively dry soil conditions to a continuous snow cover and frozen soil in the near surface. The weather after the snowfall event was consistently cold, with no snowmelt or rainfall, so soil moisture was static and the only change in gamma ray attenuation can be attributed to the accumulation of a snowpack. Wind redistribution of snow was a function of topography with transport from flat and wind exposed ridges and north west facing slopes to deposition locations in relatively wind sheltered locations on south east facing slopes. Development of transverse dunes (Filhol and Sturm, 2015) in wind-exposed locations also provided an increase in small-scale $SWE$ spatial variability. In contrast, the spring survey at this site, with the exact same flight profile as in the fall survey, observed end of the winter conditions and thus represents the accumulation and wind redistribution of several snowfall events over the winter, resulting in a generally deeper snowpacks on southeast facing lee slopes with greater spatial variability in flat areas with development of transverse, sastrugi and barchan dune snowdrifts (Filhol and Sturm, 2015). For the second season, the grassland site was surveyed at a lower altitude and slower flight



speed, with denser flight line spacing. The grassland site had greater *SWE* than that observed in the stubble field surveys and spatial variability was primarily due to relatively large snowdrift formation in the lee of fences. There was a positive relationship between vegetation height and snow depth and taller vegetation suppressed the formation of snowdrift dunes. A significant mid-winter melt event took place February 7-10, 2022 with maximum air temperatures reaching 6 °C and a 15 cm decrease in snow depth observed at a GWFO meteorological station 10 km from the study site. Snow cover remained continuous and meltwater flow through the snowpack and refreezing as a spatially discontinuous basal ice lens were observed during snow surveys.

### 2.2.2 Gamma Observations

Gamma emissions were observed with a Medusa Radiometrics MS-1000 passive gamma-ray spectrometer mounted on a Freefly AltaX UAV platform (Figure 1). Flight planning and control was done with the ALTA_QGroundControl software. Flight navigation used regular GPS signal for stubble surveys (± 5 m positioning) while navigation for grassland flights used an updated RTK system (cm level positioning). The MS-1000 utilised a 1 s integration time for gamma emissions and observed GPS, air temperature, humidity and air pressure information with an integrated sensor.

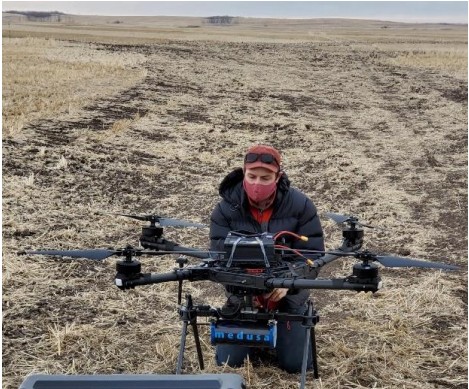

**Figure 1: Medusa MS-1000 mounted on a FreeFly AltaX prior to survey November 6, 2020. Photo credit Anders Hunter.**

In airborne applications with the spectrometer offset from the surface, airborne corrections are often implemented in order to account for the interactions of gamma rays in the air mass as well as to correct for radon and cosmic ray emissions that share this part of the electromagnetic spectrum. The Gamman software included with the MS-1000 by Medusa Radiometrics provides tools for airborne corrections with a full spectrum analysis approach. As flights were ≤ 15 m above the ground surface, where airborne corrections do not make a significant difference versus the uncertainty introduced, no airborne corrections were applied based on advice of the manufacturer. Gamman was used to perform energy stabilisation of the spectra and generate count rates ($C$) and corresponding latitude, longitude and height data at 1 second intervals. Due to data gaps in MS-1000 GPS data, the AltaX flight telemetry was used to resolve sensor trajectory. Manual alignment of the telemetry and MS-1000 GPS data was needed due to timestamp mismatches. Precision of the GPS data accessible from the AltaX telemetry logs was degraded despite RTK navigation so a 13-point rolling average was used to smooth the positioning data. The 13-point rolling average was a compromise between increased precision and alignment with the known flight path. All data at the ends of the flight lines associated



with platform slowing and turning around waypoints was removed with spatial clipping to ensure that count rate observations represented consistent flight speeds and footprint characteristics.

### 2.2.3 Validation Data

A reference dataset of $SWE$ ($SWE_{ref}$) was developed from UAV-lidar $h_s$ and snow survey $\rho_{snow}$ observations. UAV-lidar surveys quantified the spatial variability of $h_s$ at a 0.25-m spatial resolution. A Freefly AltaX UAV platform with a Riegl miniVUX2-UAV lidar was flown over the extent of the snow-covered survey areas on the same day as gamma flights. The data processing workflows to generate digital surface models (DSM) are detailed in Harder et al. (2020). The $h_s$ was computed as the difference between the snow-covered DSM and existing snow-free DSM's of the respective sites. Flights were conducted at an elevation of 110 m, with 80 m between flight lines, at a speed of 10 m s$^{-1}$. The overall $SWE_{ref}$ uncertainty ($\Delta SWE_{ref}$: mm) was propagated from the uncertainty of the observed snow density ($\Delta\rho_{snow}$) and UAV-lidar snow depth observations ($\Delta h_{s-UAV}$) as

$$\Delta SWE_{ref} = \sqrt{\frac{\sum_{i=1}^{i=n}\left(SWE_i \cdot \sqrt{\left(\frac{\Delta h_{s-UAV}}{h_{s-UAV,i}}\right)^2 + \left(\frac{\Delta\rho_{snow}}{\rho_{snow}}\right)^2}\right)^2}{n}}, \qquad (3)$$

where $i$ indexes all snow $h_{s-UAV}$ observations between 1 and $n$ (total number of observations). The $\Delta h_{s-UAV}$ was assumed to be 5 cm, a conservative value for this domain from the literature (Harder et al., 2020; Jacobs et al., 2021). For each flight, manual snow surveys collected between 12 and 60 observations of $\rho_{snow}$ with an ESC-30 snow tube (Pomeroy and Gray, 1995). Survey specific mean $\rho_{snow}$ was calculated and its uncertainty ($\Delta\rho_{snow}$) was estimated via error propagation. Assuming an $h_s$ uncertainty ($\Delta h_s$) of 1.27 cm (ruler had increments of inches) and snow mass uncertainty ($\Delta mass$) of 5% ($0.05 \cdot mass$) the uncertainty of individual $\rho_{snow}$ observations were consolidated to a survey scale $\Delta\rho_{snow}$ as:

$$\Delta\rho_{snow} = \sqrt{\frac{\sum_{i=1}^{i=n}\left(\rho_{snow,i} \cdot \sqrt{\left(\frac{\Delta h_s}{h_{s,i}}\right)^2 + \left(\frac{0.05 \cdot mass_i}{mass_i}\right)^2}\right)^2}{n}}, \qquad (4)$$

where $i$ indexes the individual $\rho_{snow}$ observations, and its constituent terms, for the respect surveys.

### 2.3 Gamma $SWE$ Processing

To relate gamma emissions observed from a moving passive sensor to a spatially distributed $SWE$ is a signal to noise and interpolation challenge. Two main factors need to be considered: the first being the temporal stability of a gamma observation, and the second the footprint it represents. At one second integration intervals, and a scintillation crystal volume of 1 L, count rates are often unstable and, based on the flight profiles employed, each observation will have overlapping footprints in longitudinal and lateral dimensions.

### 2.3.1 Count Rate Stability

To understand the temporal stability of this system, $C$ observations were analysed at start of every flight when the system was static on the ground surface. The mean $C$ for a 75 second interval was assumed to be the true $C$ of the





surface. Aggregating the 1 second $C$ with rolling means between 1 and 75 seconds simulates different integration

times. Computing the coefficient of variation (CV) for the difference in integration time mean and the 75 second mean

was used to articulate a relationship between signal stability and integration time. This provided a means to estimate

the integration period required to establish a stable $C$.

### 2.3.2 Spatial Representation

A drop-in-the-bucket (DIB) oversampling scheme was used to resolve a gridded product with minimal noise (Long et

al., 2019) as common grids are needed to compare observed and estimated $SWE$, and determine errors when varying

spatial resolution. Spatial interpolation techniques such as kriging or spline interpolation, were not implemented in

this work to avoid associated biases and artefacts and rather focus on the implications of spatial resolution and number

of individual observations aggregated. For DIB, a dense grid was generated for the respective areas of interest with

resolutions ranging between 10 and 50 m at 2.5-m intervals. For each grid resolution the mean $C$, and number of 1

second integrations included, at each grid point are computed from all points within a radius equivalent to the distance

between the centre and corner of the raster pixel. Upon computation of the respective $C$ for the various resolutions,

and snow and snow-free situations, the $C$ values were input to equation 1 to compute $SWE$. Henceforth all $SWE$

estimated from gamma observations are denoted as $SWE_{gam}$. The $SWE_{ref}$ was resampled to the respective

resolutions to allow for direct comparison with the $SWE_{gam}$.

### 2.3.3 Gamma and Lidar Data Fusion

A completely non-contact UAV based $SWE$ ($SWE_{gam-lid}$) was made by fusing high resolution $h_s$ from lidar data and

density from $SWE_{gam}$. A field-scale mean snow density ($\rho_{snow}$) was quantified from a field-scale mean gamma $SWE$

($\overline{SWE_{gam}}$) and an independent field-scale mean snow depth $\overline{(h_s)}$ from lidar as,

$$\rho_{snow} = \frac{\overline{SWE_{gam}}}{\overline{h_s}}. \tag{5}$$

The $\rho_{snow}$ in turn was reapplied to the spatially variable $h_s$ to estimate spatially distributed $SWE_{gam-lid}$ as

$$SWE_{gam-lid} = h_s \cdot \rho_{snow}. \tag{6}$$

### 3 Results

### 3.1 Snow density uncertainty

The uncertainty of $SWE_{ref}$ was compromised of observational errors associated with density and depth observations.

For the respective manual snow surveys the mean $\rho_{snow}$ and uncertainty was summarised in Table 2. No meaningful

relationships between $h_s$-$\rho_{snow}$ (Figure 1) were observed, so survey average values of $\rho_{snow}$ are deemed to be

appropriate.




**Table 2. Snow density mean and uncertainty for respective snow surveys.**

| Survey | $\rho_{snow}$ | $\Delta\rho_{snow}$ |
|---|---|---|
| Fall Stubble | 0.256 | 0.025 |
| Spring Stubble | 0.312 | 0.023 |
| Grassland | 0.249 | 0.017 |

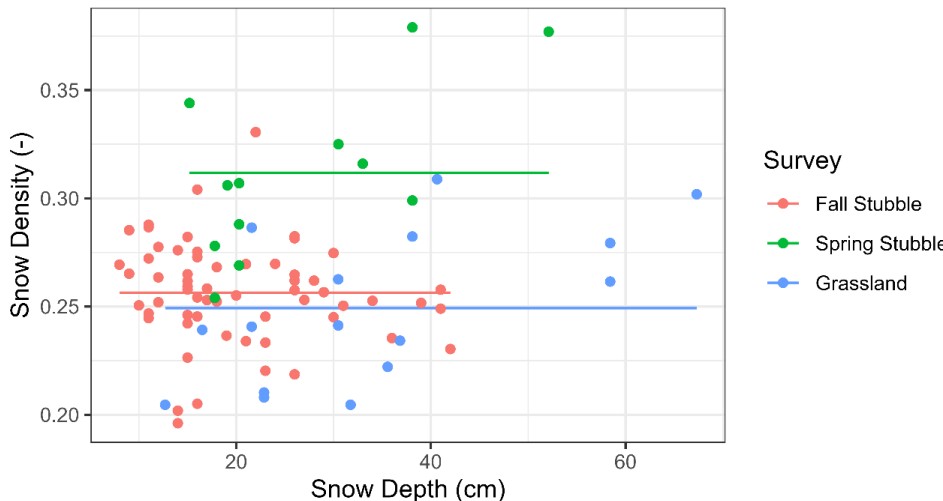

**Figure 2: Manual snow survey density (-) versus snow depth (cm) observations (top) with mean value (horizontal solid line) for respective surveys (colour).**

**3.2 Count Rate Stability**

Stable count rates are needed to ensure confidence that meaningful observations are being collected. For this, the primary factor, specific to the volume of the scintillation crystal, was the integration time. Operating the spectrometer on the ground prior to takeoff demonstrated the influence of integration time (Figure 3). By varying the integration time with application of different rolling mean windows it was evident that the CV decreases logarithmically with
integration time while mean bias was relatively stable. The longer the integration time the lower the CV. An inflection point in integration time occurs near 20 seconds where CV was between 0.01 and 0.02. Longer integration times have a decreasing rate of CV change.



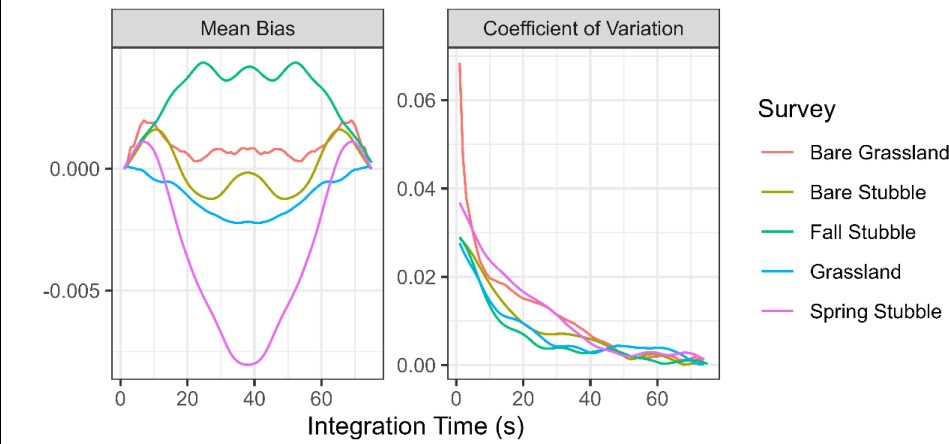

**Figure 3: Mean bias and coefficient of variation for static operation, prior to all survey flights, of UAV passive gamma ray spectrometer with varying integration time.**

### 3.3 Errors versus Spatial Resolution

The root mean square error (RSME), mean bias, and coefficient of determination ($r^2$) errors of $SWE_{gam}$ versus the resampled $SWE_{ref}$ are shown in Figure 4. The RMSE and $r^2$ improve as the spatial resolution increases while the mean bias remains static. An important dynamic was the influence of flight characteristics on survey errors. The surveys conducted at the stubble site, which had higher altitudes, wider line spacing and higher speed clearly show higher errors than the slower, lower, and narrow flight spacing of the grassland surveys. The median number of points for each raster cell for the bare and snow-covered surveys are also noted. For grassland surveys, the 22.5 m spatial resolution was associated with approximately 20 gamma observations. In contrast for stubble surveys, a spatial resolution of 35 m is required before the median number of observations reaches a similar 20 observation target. The 22.5 m resolution coincides with an inflection point for the RMSE and $r^2$ metrics for the grassland survey. The RMSE and $r^2$ values decrease between 10 and 22.5m resolutions and thereafter the rate of change slows. Variability in the grassland metrics begins to appear at the 22.5 m resolution and was explained by the overall extent of the area increasing and decreasing as pixels progressively increase in size and entire rows/columns on the edges of the extent are dropped progressively.



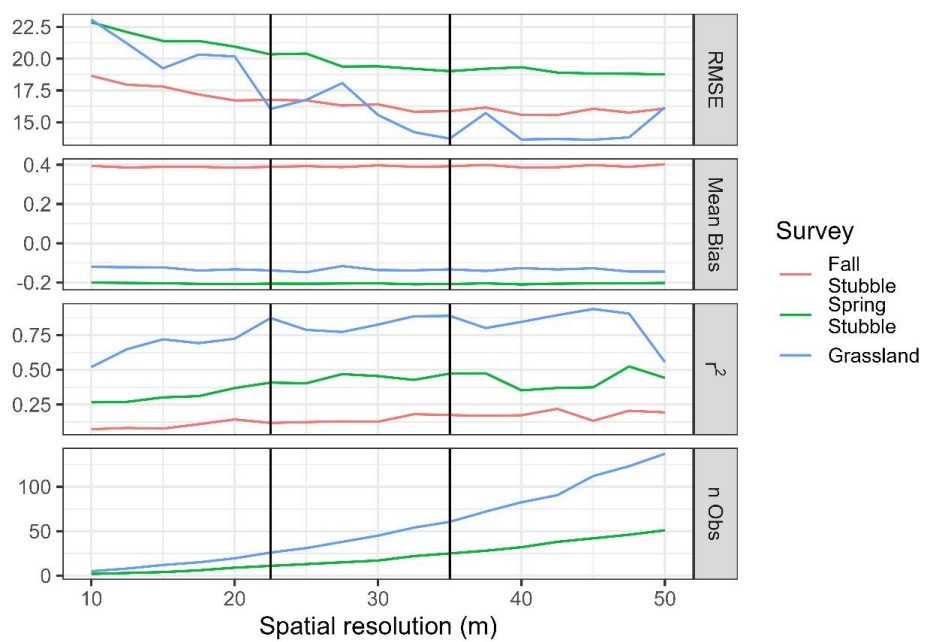

**Figure 4: The root mean square error (RMSE), mean bias, coefficient of determination (r², ), and median number of count rate observations versus raster resolution for all surveys. RMSE, mean bias, and r² are computed relative to resampled $SWE_{ref}$. The 22.5-m and 35-m spatial resolutions are highlighted by the respective vertical black lines.**

The scatter plot between the resampled $SWE_{ref}$ and $SWE_{gam}$ in Figure 5 for 22.5-m and 35-m resolutions demonstrates the high and low biases of fall and spring stubble surveys respectively. The grassland relationship was stronger with limited bias in the $SWE_{gam}$, though the variability was muted relative to the resampled $SWE_{ref}$.





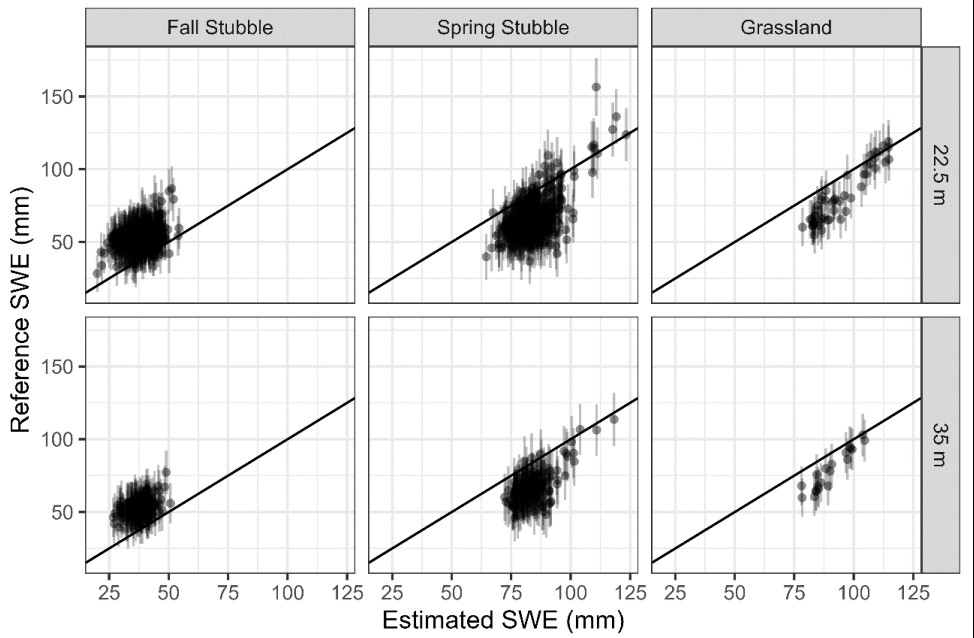

**Figure 5: UAV-lidar and snow density survey reference versus UAV-gamma estimated snow water equivalent for 22.5 and 35 m resolutions for respective surveys with 1:1 line plotted. Vertical errors bars are the propagated uncertainty of the $SWE_{ref}$.**

Comparisons of the spatial features discernible for the 22.5 m resolution $SWE_{gam}$ and $SWE_{ref}$, and in original 0.25-m resolution, visualise the ability of the technique to discern $SWE$ features (Figure 6). The negative bias of the fall stubble $SWE_{gam}$ was evident and with little spatial coherence to the resampled $SWE_{ref}$. While muted and nosier than the resampled $SWE_{ref}$ the diagonal snowdrift features in the south east of the domain was captured by the gamma in spring stubble survey. The grassland survey demonstrates the most coherence between the 22.5 m resampled $SWE_{ref}$ and $SWE_{gam}$. The snowdrifts on the north and south are evident as well as increases in $SWE$ in the depressions in the centre of the domain. Overall, the variability of the $SWE_{gam}$ was much more muted than the $SWE_{ref}$.




**Figure 6: Snow water equivalent maps at 22.5 m resolution from UAV-gamma technique (top), 22.5 m resampled UAV-lidar and snow density survey reference, $SWE_{ref}$ (middle) and 0.25 m $SWE_{ref}$.**



### 3.4 Statistical Properties of $SWE$ Distributions

Statistical properties of the $SWE$ distributions, specifically the mean and CV of $SWE$ for the respective survey areas
were computed from the 22.5-m resolution $SWE_{gam}$ and $SWE_{ref}$, as well as the 0.25-m resolution $SWE_{ref}$ (Table
3). The mean $SWE_{ref}$ was similar for the 22.5 and 0.25 m resampling as a common survey area was used. The mean
$SWE$'s provide coarse scale metrics analogous to traditional airborne gamma survey metrics. The mean $SWE_{gam}$ for
grassland was within the uncertainty bound of the $SWE_{ref}$ (from Eq. 3) at 22.5-m and 0.25-m resolutions. For fall
and spring stubble the mean $SWE_{gam}$, except for fall 22.5 m resolution $SWE_{ref}$, was outside of the uncertainty range.
The smaller magnitude of $SWE$, and larger uncertainty, for stubble surveys reduces confidence in these surveys. The
CV of the 0.25-m resolution $SWE_{ref}$ was the highest of all the surveys (ranging between 0.3 and 0.43). The resampled
22.5 m $SWE_{ref}$ the CV drops (between 0.14 and 0.29). Other than Fall stubble, which had a slighter higher CV for
$SWE_{gam}$ at 0.15 versus $SWE_{ref}$ at 0.14, the $SWE_{gam}$ was lower than the 22.5 m $SWE_{ref}$, ranging between 0.10 and
0.15.

**Table 3. Snow water equivalent site summary statistics for gamma (22.5 m) and lidar based (22.5 m and 0.25 m) resolution $SWE$**

| Survey | $SWE_{gam}$ (22.5 m) | | $SWE_{ref}$ (22.5 m) | | | $SWE_{ref}$ (0.25 m) | | |
| --- | --- | --- | --- | --- | --- | --- | --- | --- |
| | Mean | CV | Mean | CV | Uncertainty | Mean | CV | Uncertainty |
| Fall Stubble | 38.1 | 0.15 | 52.9 | 0.14 | 13.9 | 53.0 | 0.30 | 13.9 |
| Spring Stubble | 83.9 | 0.10 | 66.6 | 0.21 | 16.5 | 66.7 | 0.36 | 16.5 |
| Grassland | 94.3 | 0.12 | 81.2 | 0.23 | 13.8 | 81.8 | 0.43 | 13.9 |

To compare the statistical distributions of the different SWE representations, density plots are shown in Figure 7. All
22.5-m resolution data had lower CVs than the 0.25-m resolution and were also lower than the reference distribution.
Resampling of the 0.25-m resolution observations to coarser scales meant similar mean values but reduced variability.
From Table 3, the CV of 22.5 m $SWE_{ref}$ is 53% of the 0.25 m $SWE_{ref}$. The $SWE_{gam}$ means are higher (Grassland
+12.5 mm and Spring Stubble +17.2 mm) or lower (Fall Stubble -14.9 mm) than the 0.25m $SWE_{ref}$ with the greatest
departures for the stubble sites. Only the Grassland $SWE_{gam}$ was within the uncertainty bounds of the corresponding
0.25 m $SWE_{ref}$. Variability of $SWE_{gam}$ was also lower with the mean CV 34% of the corresponding 0.25 m $SWE_{ref}$
areas. The grassland $SWE_{gam}$ demonstrates greater variability than the stubble surveys. The grassland $SWE$
distribution shows a bimodal distribution that was evident for all resolutions and observation techniques. Regardless
of technique utilized it was apparent that the 22.5-m resolution data struggles to accurately capture the
statistical/spatial variability of the 0.25 m $SWE_{ref}$ data.





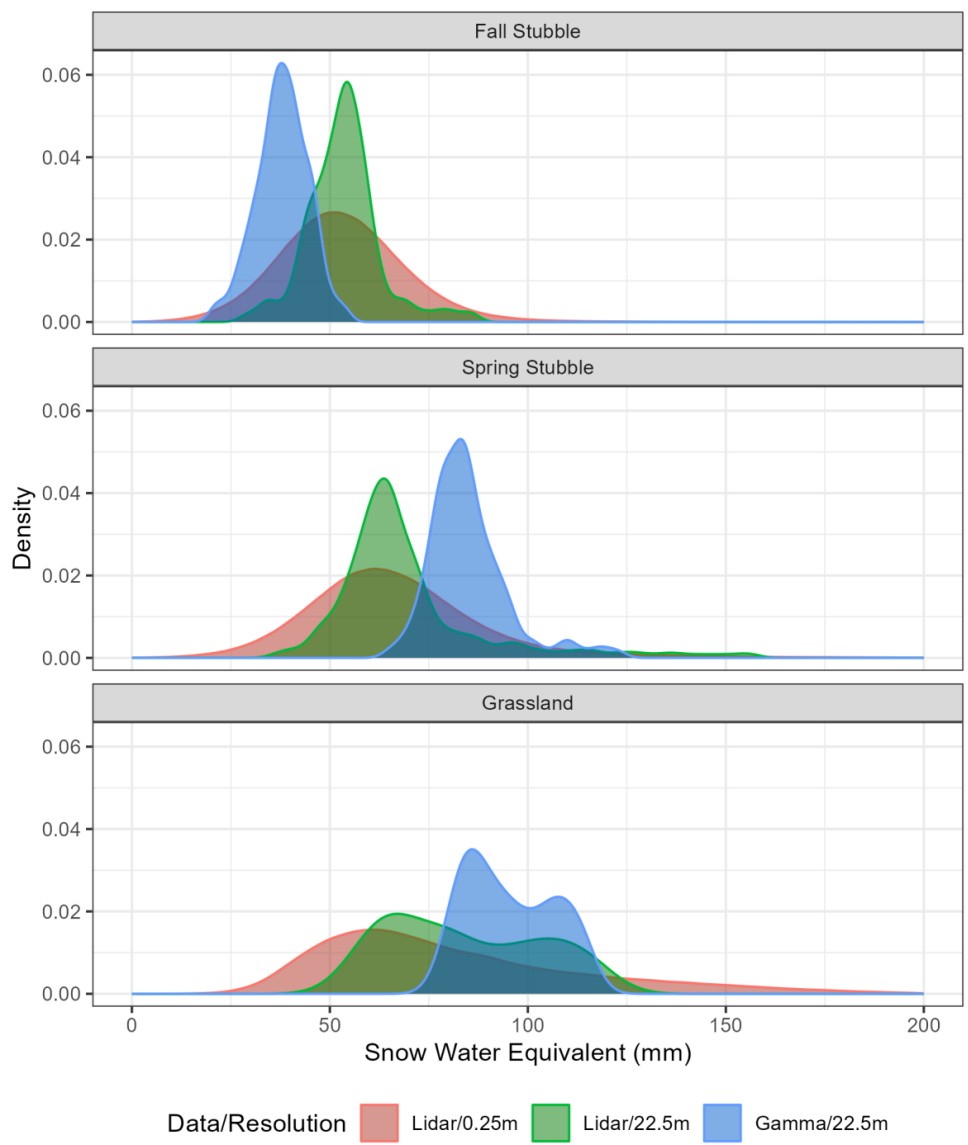

 **Figure 7: Density plots of snow water equivalent for the different surveys (rows) and estimation method/resolution (colour).**

**3.5 High Resolution $SWE$ from Gamma-Lidar Fusion**

Combing lidar-derived $h_s$ and $SWE_{gam}$ observations of grassland demonstrates a workflow to estimate $SWE$ at a 0.25-m resolution using completely remote sensing methods that require no manual snow survey (Figure 8).The average value of $SWE_{gam-lid}$ was 95 mm while the corresponding $SWE_{ref}$ (Table 3) was 82 mm and the RMSE

between the two was 14.3 mm. The difference map in Figure 8 between $SWE_{ref}$ and $SWE_{gam-lid}$ demonstrates that the fusion approach overestimated $SWE$ as function of snow depth owing to a constant $P_{snow}$ being applied.





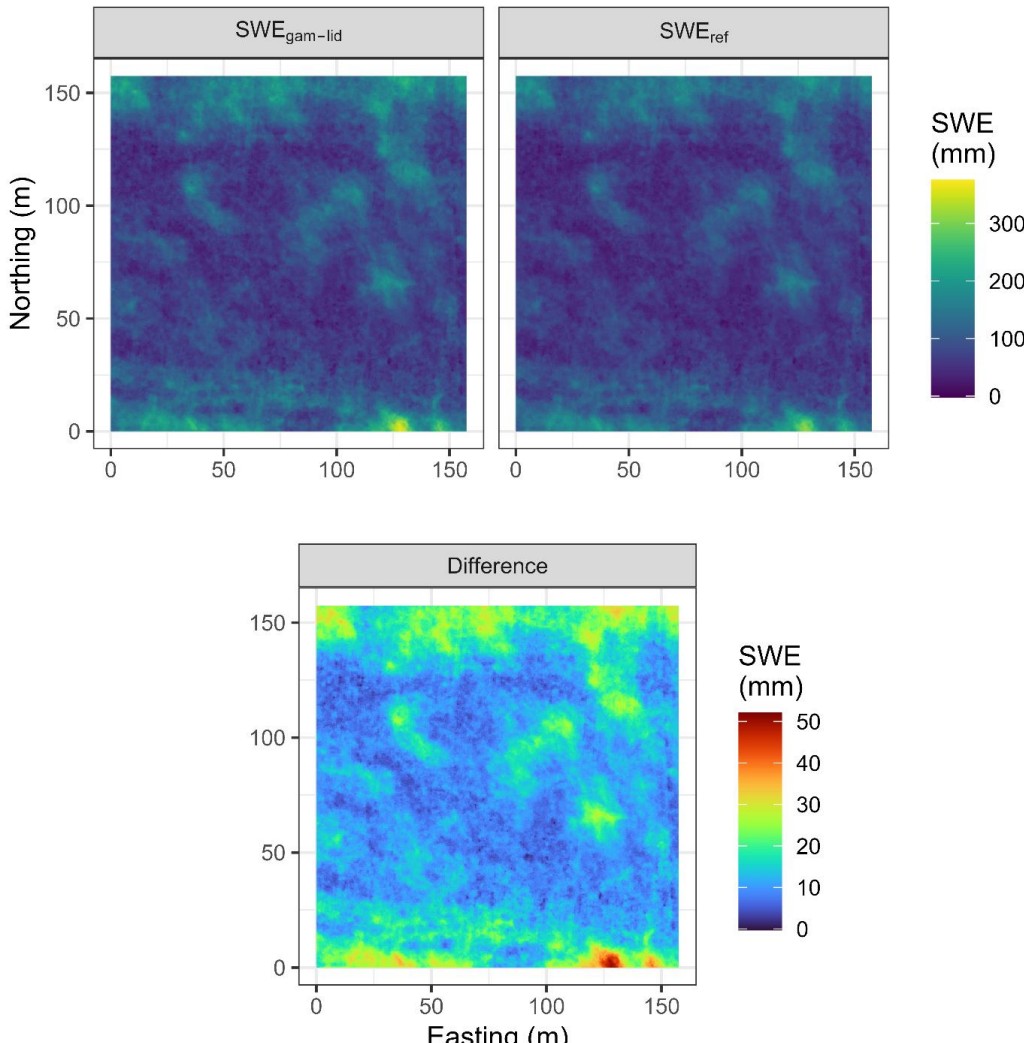

**Figure 8: High resolution (0.25 m) snow water equivalent ($SWE$) estimated from UAV-lidar snow depth and UAV-gamma $SWE$ fusion (top left) versus reference UAV-lidar and manual snow survey density $SWE$ (top right) and their difference (bottom) for the grassland site.**

## 4 Discussion

### 4.1 Accuracy, Spatial resolution, Flight characteristics, and Snowpack Scaling interactions

Relating the error metrics between spatial resolution and respective flights profiles demonstrates the many challenges of UAV-borne gamma spectroscopy to capture $SWE$ spatial variability. Temporally integrating the spectral observations is a common approach to stabilise the gamma signal and the minimum integration time was identified as an inflection point at 20 seconds in Figure 3. 10 to 100 m length scales are typically needed to capture the spatial



variability of a prairie snow cover, with a +30 m "fractal cutoff" length scale reported to overcome autocorrelation effects on flat open Canadian Prairie fields (Shook and Gray, 1996). For UAV operations, a 20 second integration time created long and narrow elliptical footprints (i.e., grassland flight footprints were approximately 15 m wide and 320     95 m long) that exceeded the 30 m fractal cutoff reported for analogous snowfields (Shook and Gray, 1996). To avoid elliptical footprints, a DIB approach to meet the integration threshold was applied that resulted in similar areal extents but circular shapes (grassland flights give approximate footprints with a radius of 21.6 m). The stabilisation of the relation between error metrics and resolution occurred at 22.5 m and 35 m resolutions for grassland and stubble surveys respectively, which aligns with the integration time threshold. Error stabilisation for grassland at 22.5 m was associated 325     with a 16.0 mm RMSE, -0.14 mm bias and 0.87 $r^2$. For the 35 m interval stubble surveys, the RMSE's were similar (15.9 mm and 19.0 mm for fall and spring stubble, respectively) but the larger biases (0.36 mm and -0.24 mm for fall and spring stubble, respectively) and lower $r^2$ (0.17 and 0.47 for fall and spring stubble respectively) imply that variability was not being captured as well. These interactions demonstrate the scaling challenges of trying to extract spatial information on $SWE$ from UAV-gamma. The slower, lower and denser flights lines over the grassland reduced 330     the footprints enough to begin to converge on the underlying $SWE$ variability while stubble flight footprints and $SWE$ variability did not align. The flight characteristics required to meet specific resolution objectives will be sensor specific and a proposed approach to guide flight planning best practices is articulated in the appendix.

### 4.2 Non-Contact High Resolution $SWE$ with Sensor Fusion

An ongoing need for snow hydrology is to be able to remotely sense wind-redistributed snowpack $SWE$ at high 335     resolutions without resorting to supplementary surface observations. The large gamma footprints relative to snowpack scale variability, as discussed, challenge the use of gamma techniques alone to directly measure $SWE$ spatial variability. Notwithstanding, UAV-borne gamma spectroscopy does have value in fusion with high-resolution snow depth estimates from lidar, or possibly other approaches such as UAV-based structure from motion, providing opportunities for this tool to advance remote snowpack measurement and mapping.

The overestimation of $SWE$ can be partly explained by a melt event earlier in the winter. Shallow snow, with less cold content to buffer a positive energy balance and lower liquid water holding capacity to absorb snowmelt, experience relatively greater melt and snowpack outflow than the deeper drifts (Gray and Landine, 1988; Fernández, 1998; Pomeroy et al., 1998). The $SWE_{ref}$ was based upon a snow depth derived from a surface difference, and so will not reliably measure snowpack changes due to meltwater redistribution and refreezing. In contrast the $SWE_{gam}$ will 345     still be influenced by the presence of this refrozen water. The complexity of snow mid-winter melt snow processes, and the inability to map the accumulation, redistribution, and refreezing of the meltwater non-destructively and independently at the snow-soil interface, complicates validation of $SWE_{gam-lid}$ with respect to the depth based $SWE_{ref}$.

The ability to discriminate between water or ice stored in the snowpack and that which infiltrated or runoff, can 350     be important depending upon the research question or application. In shallow snowpacks such as found in the Canadian Prairies, midwinter melts can be responsible for hydrologically significant changes in the snow and snow-soil interface, UAV-gamma is not likely to not observe changes in SWE plus near surface soil water/ice mass. This creates





challenges in situations where SWE estimates are important but also creates opportunities. For instance, quantifying
the total water change in the snow and near surface water/ice is incredibly valuable for estimating end of winter
changes in water stored in soil and snowpack. The total water input available from midwinter melts and snow
accumulation for soil moisture recharge and runoff is critical to inform agricultural production potential (Harder et
al., 2019) and spring freshet (He et al., 2023) in this sub-humid environment. Thus, a method that quantifies the net
input of water to soil water balance and runoff potential, that an end of winter snow specific observation would miss,
has great value. Application of this $SWE_{gam-lid}$ approach elsewhere will need to be cognizant of the saturation limits
of gamma methods for changes in water present in both the snowpack and near surface and should not be applied to
deep snow environments without further testing.

**4.3 Spatial variability of snow**

The spatial variability of SWE can be described statistically (Steppuhn and Dyck, 1974) which permits calculation of
snow cover depletion curves (Pomeroy et al., 1998). Specifically, a two-parameter log-normal distribution is often
observed in shallow snow situations (DeBeer and Pomeroy, 2010; Essery and Pomeroy, 2004a; Faria et al. 1999;
Janowicz et al., 2003; Shook and Gray, 1996), and provides a theoretical basis to predict snow cover depletion.
Development of tools that can reliably estimate these distribution parameters from remote sensing, such as with the
UAV-based sensors assessed herein, would greatly improve the capacity to understand and model prairie snowmelt
dynamics. The large differences between the $SWE$ distribution in response to resolution and lidar or gamma-based
techniques (Figure 7) complicate the ability to parametrise statistical representation of SWE directly from gamma
observations. The log-normal approaches were originally developed from snow survey datasets in uniform landscape
units (Steppuhn, 1975). DeBeer and Pomeroy (2010) needed to consider landscape classes, based on topographic
position and shallow versus deep snow classes, in order to fit observations, in a small mountain basin, to a log-normal
distribution. Faria et al., (2000) found deviations from the log-normal distribution due to inhomogeneous melt in a
boreal forest. The more detailed and spatially distributed information now available from UAV-based sensors, which
capture a wide range of landscape features equally well, provide more information than simple statistical approaches
within landscape units can summarise. This work highlights the need to consider how high-resolution distributed snow
information in the prairies may need to be discretized to meet the assumptions of log-normal statistical approaches or
if different statistical approaches are needed to estimate snow cover depletion over field scales.

**4.4 Limitations**

A key advantage of UAV versus airborne deployments is that the low and slow operations with precise positioning
will allow precise spatial co-registration of gamma emission observations from different observation intervals.
Challenges in the data processing of the observations were due to gaps and low precision in the available positioning
data. Both the uncertainty of GPS positioning for survey data <3 m in addition to the unquantified difference between
flight lines associated with the snow-free and snow-covered flights contribute differences that complicate absolute
positioning and consequently the collocation of observations between flights, and how they relate to the absolute
position of surface features. The footprints of individual observations with these flight profiles are greater than the





uncertainties associated with standalone GPS observations and are not expected to have a significant influence on results presented herein. Conducting UAV operations at lower altitudes or from ground based mobile operations will require more precise absolute spatial positioning to take advantage of smaller footprints.

The airborne, radon, and cosmic corrections often implemented with passive gamma spectroscopy were not implemented here. The near surface deployment of the sensor meant corrections would have a minimal influence on count rates. Identical flight profiles and relative altitudes imply that airborne corrections should provide the same magnitude of correction between surveys and as the $SWE$ estimation is based on a ratio between snow-covered and bare surface emissions differences that would be removed through this normalisation. Radon concentrations in the atmosphere vary over time and may be a source of uncertainty. Future work will need to evaluate this assumption and test the influence of airborne and radon corrections.

The attenuation relationship to relate $SWE$ to emissions used here was based on total gamma count rates. This differs from the equation used in the NOAA program which takes advantage of spectral information to compute a $SWE$ from total counts as well as radioisotope specific emissions that differ in their response to water attenuation in an empirical approach (Tuttle et al., 2018). To avoid the empirical aspects of these derived constants, the generic attenuation was used here, but further work may be benefit from evaluating the $SWE$ attenuation with respect to specific radioisotopes.

A challenge of this approach was capturing the variability of $SWE$ which may be a consequence of gamma emission mixing within the footprint. The $SWE_{ref}$ quantifies isolated drifts that do exceed the 300 mm $SWE$ that is the upper limit of $SWE$ detection in airborne applications. Aggregation to 22.5 m resolution in which portions of the snowpack can have $SWE > 300$ mm implies integrating observations across a large footprint that will under-sample the high $SWE$ locations. Further refinements of the footprint with nearer surface flight altitudes are needed to test this feedback.

Geo-statistical interpolation techniques are the typical approach to translate irregular point observations to regularised grids. Such methods were avoided in this analysis as the interplay between integration intervals and spatial resolutions, a defining feature of passive radiometric signal to noise challenges, needed direct consideration. Interpolation techniques all have respective strengths and weaknesses, and here statistical artefacts were avoided. Opportunities to address the signal-to-noise challenges may reside in applying interpolation techniques to further refine these results.

To the authors knowledge there have been no other UAV-borne gamma spectrometer observations of $SWE$ and this work is the first to articulate the challenges associated with using differential gamma emissions to try and resolve the spatial variability of $SWE$. Many future research opportunities exist to refine $SWE_{gam}$ estimates from improving spatial resolution and precision, evaluating airborne corrections, assessing value of gamma spectral information versus bulk count rates, testing the upper limit of $SWE$ detection, and exploiting interpolation techniques.





## 5 Conclusions

Remotely sensing *SWE* at high resolution is an ongoing need to advance snow hydrology. Large-scale *SWE* monitoring with airborne gamma methods has a long history whilst UAV-deployable passive gamma spectrometer system are only recently coming to market. The ability to remotely sense the spatial variability of *SWE* with an UAV-based passive gamma spectrometer was assessed over two snow seasons. As with other airborne applications, UAV-gamma system was able to estimate areal average of *SWE* for a 2.5-hectare study site within the uncertainty (±13.9 mm) of a reference dataset based upon UAV-lidar and snow survey observations. With a drop in the bucket aggregation method to assess spatial resolution versus errors it became evident that flight profile characteristics exert significant controls on the ability to resolve the spatial variability of *SWE*. Flight profiles in the first season of observation (5 m s$^{-1}$ velocity, 15 m altitude and 15 m line spacing) struggled to capture the underlying *SWE* variability. Updated flight profiles in the second season of observation (4 m s$^{-1}$, 8 m altitude and 8 m line spacing) demonstrated an improved ability to quantify the spatially variability of *SWE* down to 22.5 resolution (RMSE: ±16 mm, r$^2$: 0.87). Clear challenges remain in capturing *SWE* variability with the flight profiles tested but do have value in informing best practices moving forward. A fusion of gamma based *SWE* and independent datasets of UAV-lidar derived snow height was identified as an approach to remotely sense *SWE* at a high (0.25 m) resolution with an RMSE of ±14.3mm with respect to the reference *SWE* dataset. Ongoing work is still needed to evaluate the ability to resolve *SWE* at even lower and slower flight profiles which will introduce higher navigation precision demands. This work demonstrates some of the challenges of UAV-based gamma *SWE* but also articulates the opportunities available to improve remote sensing of the spatial variability of *SWE* for research and operational data collection applications.

## Appendix: Flight Planning Best Practices for UAV-based gamma SWE Observations

Balancing *SWE* observation resolution and UAV platform limitations is the main challenge to employing UAV-based gamma methods to quantify the spatial variability of *SWE*. Variations in flight line spacing, altitude, and velocity influence the scale of resolvable features and flight planning best practises to inform future operations can be gleaned from this experience. Generally, two thirds of gamma counts originate from a footprint area twice the altitude in width, and twice the altitude plus distance travelled in length (Ward, 1981). Based on flight profiles this means the approximate footprints for stubble profiles are 1050 m$^2$ (30m resolution) and for grassland profiles are 320 m$^2$ (16m resolution). The relationship between flight altitude, line spacing, and velocity and resolution associated with a 20 second integration time is simulated applying the (Ward, 1981) footprint approximation in a drop in the bucket (*DIB*) approach (Figure A1). The simulated resolutions range from 4.5 m with a flight profile with a 1 m s$^{-1}$ velocity, 1 m altitude, and 1 m altitude to a 65 m resolution with a flight profile with a 10 m s$^{-1}$ velocity, altitude of 15 m and line spacing of 15 m. The stubble flight profile aligns with a 53 m footprint resolution which demonstrates the challenges the error versus resolution patterns demonstrated in Sec 4.3 which had high errors up to the maximum 50m resolution tested. In contrast the grassland profile aligns with a 30 m footprint resolution which aligns with the plateauing of errors in the 20-30 m resolution range (Figure 4). The relative implications of flight profiles on resolvable features can be estimated from the interaction visualised in Figure A1. In the Canadian prairie context where, in uniform land



scape classes, sampling needs to span lengths scales between 30 to 100 meters to capture the spatial variability of *SWE* (Shook and Gray, 1996), it is apparent that the grassland flight profile employed is on the edge of capturing *SWE* variability appropriately. Further tests of lower, slower and closer flight lines are needed. At altitudes approaching 1 m hardware demands increase as real-time terrain following guidance systems, and RTK precision is 460  needed for navigation and position logging. The system employed in this study did not have these features and so these profiles could not be tested. The influence of atmospheric attenuation will vary with altitude and is not considered in this conceptual flight-profile versus resolution simulation.

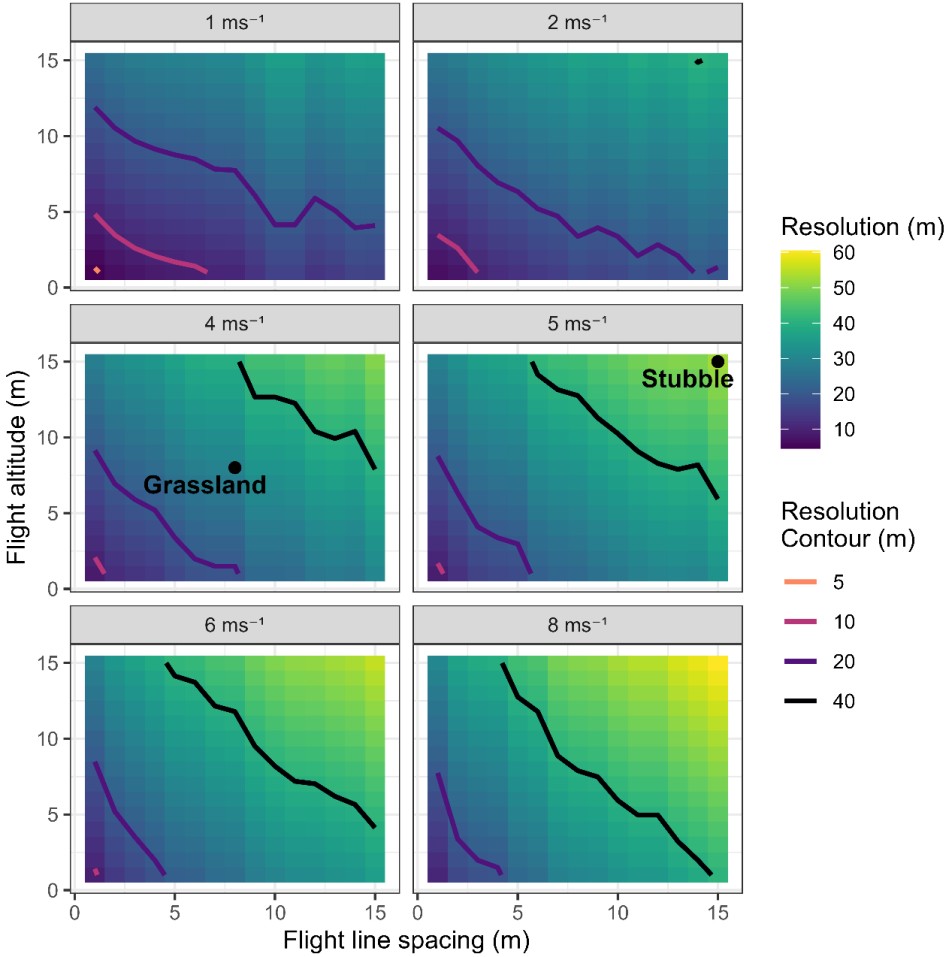

**Figure A1: Relationship between flight altitude (vertical axis), line spacing (horizontal axis), and platform velocity (panels)** 465  **versus estimated resolution (fill color) for a 20 second integration time. Contour lines of 5, 10, 20 and 40m resolutions and the points corresponding to the stubble and grassland flight profiles are plotted.**



*Code Availability:* No unique code was developed/utilized in the preparation of this manuscript.

        *Data Availability:* The underlying datasets (snow survey observations, lidar snow depth maps, gamma count rates,
        and snow water equivalent maps (reference (from lidar and observed snow density), gamma, and lidar-gamma fusion)
        are in the process of being submitted to the Federated Research Data Repository (https://www.frdr-dfdr.ca/repo/) and
a DOI is forthcoming.

        *Author Contributions:* PH, WH, and JP defined the research objectives, PH and WH performed the fieldwork, and PH
        completed the data analysis and interpretation, and manuscript preparation. All authors contributed to discussions and
        revisions of the manuscript.

        *Competing Interests.* The authors declare no competing interests.

        *Acknowledgements*. Funding for this work comes from the Canada First Research Excellence Fund through the Global
        Water Futures programme, Canadian Foundation for Innovation, Western Economic Diversification and the Canada
Research Chairs programme. Field and technical assistance from Bruce Johnson, Anders Hunter, Alistair Wallace,
        and Medusa Radiometrics is gratefully acknowledged.

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
