# Peer review of "Measuring prairie snow water equivalent with combined UAVborne gamma spectrometry and lidar"

_EGUsphere, 2023_

## Referee Comment (RC1)

This study evaluated the ability to remotely sense the spatial variability of snow water equivalent (SWE) with a UAV-based passive gamma spectrometer, focusing on shallow prairie snowpacks over two snow seasons in Saskatchewan, Canada. The UAV-gamma technique successfully captured the areal mean and spatial variability of SWE with slower, lower, and denser flight lines in the second season. However, the first-year survey results showed limited ability to observe the average and spatial variability of SWE. The study also tested a combination of UAV-based gamma and UAV-based lidar to develop a high-resolution SWE map, showing improved results.

To the best of my knowledge, this study is the first to attempt testing the UAV-based passive gamma-ray technique to quantify SWE. The manuscript provides useful practical guidance for the snow remote sensing community, especially for those interested in applying and advancing the technique in other environments. The challenges of UAV-based gamma SWE are discussed, articulating the opportunities available to improve remote sensing of the spatial variability of SWE for data collection applications.

Generally, with this new UAV-based gamma approach, the study addressed clear scientific questions, presented in a well-structured way. I believe this will make progress beyond the current scientific understanding of a passive gamma-ray approach for SWE estimation for The Cryosphere community. In my opinion, however, there are a few things, particularly related to the gamma SWE retrieval approach, that need to be addressed before publication. I have provided a few suggestions below that may help to the improvement of the manuscript.

**General comments**

1. My understanding is that the MS-1000 gamma spectrometer measures gamma count rates of 40K, 238U, 232Th, and 137Cs radionuclides. When the authors apply Eq 2 to calculate SWE, how did they calculate the count rates ($c_{bare}$ and $c_{snow}$) from the individual gamma elements? If the authors used 'total count rates' regardless of radionuclides, can the authors provide the gamma radiation spectra like Figure 12 in Offenbacher and Colbeck (1991)? That would be helpful for potential readers to better understand the passive gamma radiation technique and its attenuation effect by snowpack. I would also suggest providing spatial maps of gamma count rate for each radionuclide along with the total count rates as supplementary.
   - *Offenbacher, E. L. and Colbeck, S. C.: Remote sensing of snow covers using the gamma-ray technique (No. AD-A- 238016/0/XAB; CRREL–91-9), Cold Regions Research and Engineering Lab., Hanover, NH (United States), https://apps.dtic. mil/sti/pdfs/ADA238016.pdf, 1991.*

2. Related to Comment 1.1, it should be clearly articulated why they used total gamma count rates rather than specific radioisotope components such as 40K, which are expected to be more sensitive to better quantify the attenuation by SWE (Peck et al., 1971; Offenbacher & Colbeck 1991). I understand the authors want to avoid the empirical aspects; However, I still think some justifications would be valuable. Specifically, if possible, a simple comparison of the gamma SWE values derived from total gamma counts used here and specific gamma radioelements (e.g., previous studies were used) would be valuable for potential readers.
   - *Peck, E. L., Bissell, V. C., Jones, E. B., & Burge, D. L. (1971). Evaluation of snow water equivalent by airborne measurement of passive terrestrial gamma radiation. Water Resources Research, 7(5), 1151-1159.*

3. Regarding the overestimation of SWE in Grassland in the second winter, I agree with the discussion point (L340-348) that a melt event earlier can contribute to the SWE uncertainty. To further discuss, I think it would be helpful if a time series of soil moisture and SWE throughout the season can be provided (if the station data is available near the sites). In this context, it might also be helpful to include a time series of temperature, snow depth, and precipitation at a GWFO station in the main body or as supplementary material for a better understanding of site weather conditions before and after the surveys.

**Detailed comments**

L10-11 I suggest removing 'over shallow snowcovers' from the statement. Airborne applications using passive gamma-ray surveys have the capability to measure relatively moderate or deep snowpacks. The operational NOAA airborne gamma survey provides up to 1000 mm of SWE (https://www.nohrsc.noaa.gov/snowsurvey/historical.html; Mortimer et al., 2024).

> *Mortimer, C., Mudryk, L., Cho, E., Derksen, C., Brady, M., & Vuyvich, C. (2024). Use of multiple reference data sources to cross validate gridded snow water equivalent products over North America. EGUsphere, 2024, 1-31.*

L81-82 A recent effort to address the issue of the antecedent soil moisture change for the airborne program using SMAP soil moisture was presented in Cho et al. (2020). It would be good to mention it here.

> *Cho, E., Jacobs, J. M., Schroeder, R., Tuttle, S. E., & Olheiser, C. (2020). Improvement of operational airborne gamma radiation snow water equivalent estimates using SMAP soil moisture. Remote Sensing of Environment, 240, 111668.*

Section 2.1 It would be much more helpful if general snow conditions are provided including annual SWE (maximum and range), snow covered duration, and onset/offset dates with some literatures if available.

L118 snow (water equivalent) → SWE

L141-142 For the stubble survey, how much did you expect that the larger GPS error ranges might impact the gamma SWE estimation? Please provide some justification regarding this.

Figure 1. Have the authors considered providing some additional photos of both sites showing different surface characteristics along with the descriptions provided? I believe that would be informative for readers to better understand the sites.

L201 Should it be the equation "2"?

L205 high resolution → high spatial resolution

Figure 3 How were the mean biases calculated? What is the reference count rate to calculate mean bias at a certain integration time? Please describe it briefly. Also, why were the shapes of changes in mean biases with change in integration time different particularly between Fall Stubble and Spring Stubble? Were the changes negligible in terms of the amount of count rate?

Result 3.3 Please add the units of RMSE and mean bias (Figure 4) and the mean and uncertainty (Table 3). Also check the units throughout all figures and tables.

L426 Please include the areal mean SWE values with the uncertainty for each season.

L426 2.5-hectare → 2.5 x 106 m2

L432 22.5 m spatial resolution

L432 Include the mean SWE value along with RMSE

---

## Referee Comment (RC2)

**Review: Measuring prairie snow water equivalent with combined UAV-borne gamma spectrometry and lidar**

The manuscript describes the demonstration of a new UAV-gamma spectrometer for measuring snow water equivalent over prairie snow in Canada. The results are assessed against SWE derived from UAV-lidar and manual snow density observations. Additionally, a fully contactless UAV-gamma+lidar fusion SWE product is demonstrated. To my knowledge this is the first demonstration of a UAV-gamma spectrometer applied over snow and provides the groundwork for future studies that may want to implement this new instrument at the UAV scale. The detailed considerations for flight planning and data acquisition that are presented will certainly be helpful for future studies using this instrument. The manuscript is generally well-written and organized and is suitable for publication in The Cryosphere. Below I have outlined the general comments that should be addressed prior to publication, followed by line-by-line comments. My comments are intended to be constructive and improve the overall manuscript.

General comments:

1. The main weakness of the study is the application of the manual density measurements in the reference dataset, the gamma and lidar fusion dataset, and in the defined uncertainty for the reference dataset. Figure 3 highlights large variation in density over the surveyed areas and simply taking the average density to represent the entire area introduces bias. Since this is the basis from which the SWE map is assessed, why were the locations of the density measurements considered? Were GPS points taken for each of the density measurements? I think a spatially interpolated density map would greatly improve the robustness of the analysis and the results for both the gamma SWE and gamma + lidar SWE. As for the uncertainty in the reference SWE, equation 4 describes the error in a single density measurement but does not represent the error propagation to SWE as a result of taking the average density over the survey area. The error in any given pixel should additionally contain the spread (e.g., 95% percentile) in density measurements over the survey area relative to the mean.

2. One of the main goals of the paper is to present a new instrument for measuring gamma rays, however the detailed description of the data collected by the instrument is not shown nor described in detail. Per the manufacturers website, the MS-1000 collects data for multiple radionuclides across spectral channels. Please include full details of the data collected and how the total counts C were calculated based on the collected data. An example of the collected data (potentially as a sub-figure in Figure 1) illustrating the collected data would be a nice addition that will help the reader understand the data collected with the instrument.

3. Units are missing in many locations throughout figures and tables, that I have noted below.

**Line-by-Line Comments**

Line 18: In the abstract the "reference dataset" is mentioned without context to what it is. I recommend briefly describing it here for clarity.

Line 22: I think this statement is somewhat misleading. The gamma+lidar fusion approach did not improve the results "substantially" and in fact the average site wide SWE measurement was slightly

worse compared to the gamma SWE measurement alone. What did improve substantially is the spatial resolution of the SWE distribution.

Line 25: Resolution is better described in terms of fine and coarse. I recommend that fine/coarse replace high/low when describing resolution.

Line 82: Missing the word "of"
 "Approaches to correct for overwinter changes require independent estimates of soil moisture change.."

Line 90: include the term "gamma" in UAV spectroscopy/spectrometry.

Line 95: The terms spectroscopy and spectrometry are used interchangeably in manuscript. I recommend choosing one of these terms and use it consistently, which should probably be spectrometry because that is the term used in the title.

Line 115: change was to were.

Line 116:  "to" is not needed here.

Line 124: northwest and southeast are one word.

Line 128: snowpack should be singular in this context.

Line 152: Per major comment 1, please provide more information on how the spectral measurement was turned into count rates.

Equation 4: This equation describes the uncertainty for the density measurements, but it does not describe the uncertainty that is propagated to your spatial SWE reference dataset by using the average of the density measurements across the entire survey error. The uncertainty for any given pixel in the reference SWE map should be defined somehow based on the spread of the density measurements in Figure 2 if you decide to keep the analysis based on the average density.

Line 202-203: How was the lidar data interpolated? DIB or interpolation? Please add here.

Section 2.3.3: I don't understand why this analysis was not done spatially instead of using the average height and the average SWE from the two data products. This seems like a much more robust analysis and would make for a much stronger sensor fusion analysis.

Table 2: Add units to table. Also please add a statistic that describes the spread of the density measurements (i.e., 95 percentile, standard deviation, etc.)

Figure 2: Snow density needs units on the y-axis and in the caption, I am not familiar with any convention that uses (-) as a means to describe a density.

Line 229: Consider reminding the reader here what CV stands for. It is defined much earlier in text and I had to go back to remind myself.

Figure 3: What are the units of bias in this figure (y-axis). Could you provide some more context as to why some of these are positive and negative.

Figure 4: y-axis units missing on bias, RMSE, and n. Also is the red fall stubble line missing in the number of observations plot? If it is underlying one of the other lines please make visible by increasing the line width of the underneath line or using dashed line.

Line 256: Biases should be described as positive/negative.

263: "negative bias" used here, good!

Table 2: Table missing units.

Figure 7: Consider modifying the y-axis label so that it is clear that this doesn't refer to snow density. Also, would it make sense to add the gamma_lidar fusion distribution here?

Line 306: P should be rho.

Figure 8: Make clear which way the difference is done (ie., gam_lid – ref OR ref – gam_lid)

Line 344; ""…snowpack density changes…"'

Line 402: "be" not needed

---

## Author Comment (AC1)

Reviewer #2: Chris Donahue

We are pleased to hear that you find the manuscript well-written and suitable for publication in *The Cryosphere*, and we greatly appreciate your suggestions for improvement. We are committed to addressing each of your comments to enhance the quality and clarity of our work.

 General comments:
1       The main weakness of the study is the application of the manual density measurements in the reference dataset, the gamma and lidar fusion dataset, and in the defined uncertainty for the reference dataset. Figure 3 highlights large variation in density over the surveyed areas and simply taking the average density to represent the entire area introduces bias. Since this is the basis from which the SWE map is assessed, why were the locations of the density measurements considered? Were GPS points taken for each of the density measurements? I think a spatially interpolated density map would greatly improve the robustness of the analysis and the results for both the gamma SWE and gamma + lidar SWE. As for the uncertainty in the reference SWE, equation 4 describes the error in a single density measurement but does not represent the error propagation to SWE as a result of taking the average density over the survey area. The error in any given pixel should additionally contain the spread (e.g., 95% percentile) in density measurements over the survey area relative to the mean.

From previous work on snow density–depth relationships for prairie snow covers (shallow, windblown, and highly spatially variable) it has been demonstrated that, for snow depth < 60cm, density does not vary significantly with depth. Since the study areas are relatively small and subject to similar conditions (mainly blowing snow processes) there is little expected benefit to increasing the complexity of the density representation beyond applying a mean observed density (Shook and Gray 1994).  In contrast, the uncertainty in measurement of the density of shallow snow is such that an interpolated map will add significant noise that can be avoided with an average representation.  Not all density observations had GPS locations taken, largely due to fieldwork/logistical constraints under COVID protocols, so we do not have a consistent GPS positioning to employ.  Regarding the uncertainty in the reference SWE, we don't quite understand the comment.  The uncertainty in reference SWE (Eq3) does consider the uncertainty of the average density applied (as calculated in Equation 4).

Shook, K.R.;Gray, D.M., Determining the Snow Water Equivalent of Shallow Prairie Snowcovers., Presented at Eastern Snow Conference, (1994), 1:14

2       One of the main goals of the paper is to present a new instrument for measuring gamma rays, however the detailed description of the data collected by the instrument is not shown nor described in detail. Per the manufacturers website, the MS-1000 collects data for multiple radionuclides across spectral channels. Please include full details of the data collected and how the total counts C were calculated based on the collected data. An example of the collected data (potentially as a sub-figure in Figure 1) illustrating the collected data would be a nice addition that will help the reader understand the data collected with the instrument.

We recognize the importance of providing a comprehensive description of the data collected by the UAV-gamma spectrometer. We will include more information about the how the total counts C were calculated and add a figure to the supplement if warranted.

3       Units are missing in many locations throughout figures and tables, that I have noted below.
We acknowledge the oversight in missing units in various parts of the manuscript. We will ensure that all figures and tables have the appropriate units clearly stated in the revision.

**Line-by-Line Comments**

Line 18: In the abstract the "reference dataset" is mentioned without context to what it is. I recommend briefly describing it here for clarity.

We will clarify the reference dataset in the abstract.

Line 22: I think this statement is somewhat misleading. The gamma+lidar fusion approach did not improve the results "substantially" and in fact the average site wide SWE measurement was slightly worse compared to the gamma SWE measurement alone. What did improve substantially is the spatial resolution of the SWE distribution.

We will clarify that the improvement is in the greatly increased spatial representation of SWE.

Line 25: Resolution is better described in terms of fine and coarse. I recommend that fine/coarse replace high/low when describing resolution.

We will update the terminology.

Line 82: Missing the word "of"
"Approaches to correct for overwinter changes require independent estimates of soil moisture change.."

We will correct this.

Line 90: include the term "gamma" in UAV spectroscopy/spectrometry.

We will correct this.

Line 95: The terms spectroscopy and spectrometry are used interchangeably in manuscript. I recommend choosing one of these terms and use it consistently, which should probably be spectrometry because that is the term used in the title.

We will update the terminology to be consistent as suggested.

Line 115: change was to were.

We will correct this.

Line 116: "to" is not needed here.

We will correct this.

Line 124: northwest and southeast are one word.

We will correct this.

Line 128: snowpack should be singular in this context.

We will correct this.

Line 152: Per major comment 1, please provide more information on how the spectral measurement was turned into count rates.

We will update our description on this processing step. See also our response to reviewer 1 on this topic.

Equation 4: This equation describes the uncertainty for the density measurements, but it does not describe the uncertainty that is propagated to your spatial SWE reference dataset by using the average of the density measurements across the entire survey error. The uncertainty for any given pixel in the reference SWE map should be defined somehow based on the spread of the density measurements in Figure 2 if you decide to keep the analysis based on the average density.

We don't fully understand this comment as we are propagating the uncertainty of the average density and uncertainty of snow depth estimates from the UAV at each pixel prior to propagation to the overall average SWE observed for the respective study areas.

Line 202-203: How was the lidar data interpolated? DIB or interpolation? Please add here.

We will clarify this.  The UAV-lidar data is at a very high density (~100 pts/m$^2$) and we utilised LAStools approaches to convert the irregular point cloud to a 0.25 m gridded representation via a TIN surface fitting approach.  Rescaling from the 0.25 m base resolution to other resolutions used the mean value of the new/larger grids.

Section 2.3.3: I don't understand why this analysis was not done spatially instead of using the average height and the average SWE from the two data products. This seems like a much more robust analysis and would make for a much stronger sensor fusion analysis.

There is spatial variability in the fusion.  The spatial variability comes from the UAV-lidar snow depth (0.25 m resolution observations).  Only the density is an areal average per the approach/justification explained in the second major comment.  We will clarify this further.

Table 2: Add units to table. Also please add a statistic that describes the spread of the density measurements (i.e., 95 percentile, standard deviation, etc.)

We will update as suggested.

Figure 2: Snow density needs units on the y-axis and in the caption, I am not familiar with any convention that uses (-) as a means to describe a density.

We will update as suggested.

Line 229: Consider reminding the reader here what CV stands for. It is defined much earlier in text and I had to go back to remind myself.

We will update as suggested.

Figure 3: What are the units of bias in this figure (y-axis). Could you provide some more context as to why some of these are positive and negative.

We will be updating this figure to remove the mean bias panel as per Reviewer 1's comments.

Figure 4: y-axis units missing on bias, RMSE, and n. Also is the red fall stubble line missing in the number of observations plot? If it is underlying one of the other lines please make visible by increasing the line width of the underneath line or using dashed line.

Indeed, the fall and spring stubble lines are plotting on top of each other. We will modify this presentation as suggested.

Line 256: Biases should be described as positive/negative.

We will update as suggested.

263: "negative bias" used here, good!

Thanks

Table 2: Table missing units.

We will update as suggested.

Figure 7: Consider modifying the y-axis label so that it is clear that this doesn't refer to snow density. Also, would it make sense to add the gamma_lidar fusion distribution here?

The addition of the gamma_lidar fusion here is a great suggestion. We will update accordingly.

Line 306: P should be rho.

We will update as suggested.

Figure 8: Make clear which way the difference is done (ie., gam_lid – ref OR ref – gam_lid)

We will update as suggested.

Line 344; ""…snowpack density changes…""

We will update as suggested.

Line 402: "be" not needed

We will update as suggested.

Your detailed and insightful review has provided us with clear directions for improvement. We are confident that these revisions will strengthen the manuscript and make a valuable contribution to the field. Once again, we appreciate your valuable feedback and look forward to submitting a revised version of our manuscript.

Sincerely, Phillip Harder, Warren Helgason and John Pomeroy

---

## Author Comment (AC2)

Response to Reviews

Reviewer #1 : Eusang Cho

Thank you for your comprehensive and insightful review of our manuscript. Your comments and suggestions are invaluable in strengthening the study and ensuring the clarity and robustness of our findings. We appreciate the time and effort you have invested in evaluating our work. Our responses follow in red font.

**General comments**
1       My understanding is that the MS-1000 gamma spectrometer measures gamma count rates of 40K, 238U, 232Th, and 137Cs radionuclides. When the authors apply Eq 2 to calculate SWE, how did they calculate the count rates (c_bare and c_snow) from the individual gamma elements? If the authors used 'total count rates' regardless of radionuclides, can the authors provide the gamma radiation spectra like Figure 12 in Offenbacher and Colbeck (1991)? That would be helpful for potential readers to better understand the passive gamma radiation technique and its attenuation effect by snowpack. I would also suggest providing spatial maps of gamma count rate for each radionuclide along with the total count rates as supplementary.

*Offenbacher, E. L. and Colbeck, S. C.: Remote sensing of snow covers using the gamma-ray technique (No. AD-A- 238016/0/XAB; CRREL–91-9), Cold Regions Research and Engineering Lab., Hanover, NH (United States), https://apps.dtic. mil/sti/pdfs/ADA238016.pdf, 1991.*

We agree that aligning with some of the existing SWE-gamma work that focuses on the sensitivities associated with different specific radionuclides (like in Offenbacher and Colbeck (1991)) would be useful. We did attempt to implement those approaches early in the project but made the decision to focus on the total count rates, as they were shown to be more robust overall. The MS-1000 unit provides the means to isolate the emissions from specific radionuclides; however, it does this through interpretation/integration of the spectra.  In the mobile application described, using high frequency (1 second) integrations of the spectra, we found that there was an unreasonable amount of noise versus signal when looking at the SWE with respect to specific radionuclides.  Additionally, it was not possible to perfectly align (spatially) the point scale gamma spectra over snow versus over bare ground. Thus, we felt it more appropriate to utilise the total count rate approach, which has had well documented usage (Carroll, 2001), instead of the specific radionuclide approach. We will add more detailed justification to the manuscript for this approach, along with spatial maps of gamma total count rates as supplementary material as suggested.

Carroll, T.: Airborne Gamma Radiation Snow Survey Program: A User's Guide. Version 5.0, National Operation Hydrologic Remote Sensing Center., 2001.

2    Related to Comment 1.1, it should be clearly articulated why they used total gamma count rates rather than specific radioisotope components such as 40K, which are expected to be more sensitive to better quantify the attenuation by SWE (Peck et al., 1971; Offenbacher & Colbeck 1991). I understand the authors want to avoid the empirical aspects; However, I still think some justifications would be valuable. Specifically, if possible, a simple comparison of the gamma SWE values derived from total gamma counts used here and specific gamma radioelements (e.g., previous studies were used) would be valuable for potential readers.

*Peck, E. L., Bissell, V. C., Jones, E. B., & Burge, D. L. (1971). Evaluation of snow water equivalent by airborne measurement of passive terrestrial gamma radiation. Water Resources Research, 7(5), 1151-1159.*

You raise an important point about the decision to use total gamma count rates rather than specific radioisotope components. We will provide a clearer justification for this approach inline with the previous

comment, incorporating a comparison of gamma SWE values derived from total gamma counts and specific gamma radioelements, as you suggested in the supplement. This comparison will help to contextualize our approach within the broader field of study.

3    Regarding the overestimation of SWE in Grassland in the second winter, I agree with the discussion point (L340-348) that a melt event earlier can contribute to the SWE uncertainty. To further discuss, I think it would be helpful if a time series of soil moisture and SWE throughout the season can be provided (if the station data is available near the sites). In this context, it might also be helpful to include a time series of temperature, snow depth, and precipitation at a GWFO station in the main body or as supplementary material for a better understanding of site weather conditions before and after the surveys.

We understand the need to elaborate on the overestimation of SWE in the second winter. We will provide meteorological data from a nearby site (<10 km away) which will help illustrate the mid-winter melt events. Unfortunately we did not observe the soil moisture conditions at this particular grassland site during the study period.

**Detailed comments**

L10-11 I suggest removing 'over shallow snowcovers' from the statement. Airborne applications using passive gamma-ray surveys have the capability to measure relatively moderate or deep snowpacks. The operational NOAA airborne gamma survey provides up to 1000 mm of SWE (https://www.nohrsc.noaa.gov/snowsurvey/historical.html; Mortimer et al., 2024).
*Mortimer, C., Mudryk, L., Cho, E., Derksen, C., Brady, M., & Vuyvich, C. (2024). Use of multiple reference data sources to cross validate gridded snow water equivalent products over North America. EGUsphere, 2024, 1-31.*

We will revise this statement to reflect the broader capabilities of airborne applications using passive gamma-ray surveys.

L81-82 A recent effort to address the issue of the antecedent soil moisture change for the airborne program using SMAP soil moisture was presented in Cho et al. (2020). It would be good to mention it here.
*Cho, E., Jacobs, J. M., Schroeder, R., Tuttle, S. E., & Olheiser, C. (2020). Improvement of operational airborne gamma radiation snow water equivalent estimates using SMAP soil moisture. Remote Sensing of Environment, 240, 111668.*

Thank you for pointing out the study by Cho et al. (2020). We will mention this work to provide context on addressing soil moisture changes.

Section 2.1 It would be much more helpful if general snow conditions are provided including annual SWE (maximum and range), snow covered duration, and onset/offset dates with some literatures if available.

We will include general snow conditions such as annual SWE, snow-covered duration, and onset/offset dates, as supported by literature.

L118 snow (water equivalent) → SWE

We will correct.

L141-142 For the stubble survey, how much did you expect that the larger GPS error ranges might impact the gamma SWE estimation? Please provide some justification regarding this.

We don't expect the larger GPS error for the stubble survey would have had a significant impact because this site was surveyed at a higher altitude (therefore larger footprint) – thus the GPS error would be partially compensated for by the coarser resolution. We will articulate this in the edited manuscript.

Figure 1. Have the authors considered providing some additional photos of both sites showing different surface characteristics along with the descriptions provided? I believe that would be informative for readers to better understand the sites.

We will consider adding additional photos for Figure 1 to provide this context.

L201 Should it be the equation "2"?

We will correct this.

L205 high resolution → high spatial resolution

We will correct this.

Figure 3 How were the mean biases calculated? What is the reference count rate to calculate mean bias at a certain integration time? Please describe it briefly. Also, why were the shapes of changes in mean biases with change in integration time different particularly between Fall Stubble and Spring Stubble? Were the changes negligible in terms of the amount of count rate?

We will provide a better explanation of the calculations used to create Figure 3. The mean bias for the static test is taken as the difference between the average count rate observed over the whole of the 75 second interval and the mean bias for the survey flights were taken as the difference from this reference and the average of the mean of the different integration times. There are some interesting contrasts in the shapes of the different mean biases shown, and upon reconsideration we feel that this approach does not communicate the signal stability/noise that we initially had in mind. In contrast, the CV of the integration time (right panel) clearly expresses the SNR to integration time relationship much more clearly and we will remove the mean bias (left panel).

Result 3.3 Please add the units of RMSE and mean bias (Figure 4) and the mean and uncertainty (Table 3). Also check the units throughout all figures and tables.

Units for RMSE, mean bias, and mean SWE values will be added and checked throughout all figures and tables.

L426 Please include the areal mean SWE values with the uncertainty for each season.
L426 2.5-hectare → 2.5 x 106 m2
L432 22.5 m spatial resolution
L432 Include the mean SWE value along with RMSE

L118, L201, L205, L426, L432: These specific line comments are noted and will be addressed accordingly.

We are committed to addressing each of your points thoroughly to ensure the manuscript meets the high standards of the journal and contributes significantly to the field of snow remote sensing. Once again, we appreciate your constructive feedback and look forward to improving our manuscript with your suggestions.

Sincerely, Phillip Harder, Warren Helgason and John Pomeroy

---

## Author Response (AR1)

Response to Reviews

We have changed the color scheme of the figures to be more sensitive to colorblind viewers

Reviewer #1 : Eusang Cho

**General comments**

1       My understanding is that the MS-1000 gamma spectrometer measures gamma count rates of 40K, 238U, 232Th, and 137Cs radionuclides. When the authors apply Eq 2 to calculate SWE, how did they calculate the count rates (c_bare and c_snow) from the individual gamma elements? If the authors used 'total count rates' regardless of radionuclides, can the authors provide the gamma radiation spectra like Figure 12 in Offenbacher and Colbeck (1991)? That would be helpful for potential readers to better understand the passive gamma radiation technique and its attenuation effect by snowpack. I would also suggest providing spatial maps of gamma count rate for each radionuclide along with the total count rates as supplementary.

*Offenbacher, E. L. and Colbeck, S. C.: Remote sensing of snow covers using the gamma-ray technique (No. AD-A- 238016/0/XAB; CRREL–91-9), Cold Regions Research and Engineering Lab., Hanover, NH (United States), https://apps.dtic. mil/sti/pdfs/ADA238016.pdf, 1991.*

We agree that aligning with some of the existing SWE-gamma work that focuses on the sensitivities associated with different specific radionuclides (like in Offenbacher and Colbeck (1991)) would be useful. We did attempt to implement those approaches early in the project but made the decision to focus on the total count rates, as they were shown to be more robust overall. The MS-1000 unit provides the means to isolate the emissions from specific radionuclides; however, it does this through interpretation/integration of the spectra.  In the mobile application described, using high frequency (1 second) integrations of the spectra, we found that there was an unreasonable amount of noise versus signal when looking at the SWE with respect to specific radionuclides.  Additionally, it was not possible to perfectly align (spatially) the point scale gamma spectra over snow versus over bare ground. Thus, we felt it more appropriate to utilise the total count rate approach, which has had well documented usage (Carroll, 2001), instead of the specific radionuclide approach. We added more detailed justification to the manuscript for this approach, along with spatial maps of gamma total count rates as supplementary material as suggested in Appendix A.

Carroll, T.: Airborne Gamma Radiation Snow Survey Program: A User's Guide. Version 5.0, National Operation Hydrologic Remote Sensing Center., 2001.

2    Related to Comment 1.1, it should be clearly articulated why they used total gamma count rates rather than specific radioisotope components such as 40K, which are expected to be more sensitive to better quantify the attenuation by SWE (Peck et al., 1971; Offenbacher & Colbeck 1991). I understand the authors want to avoid the empirical aspects; However, I still think some justifications would be valuable. Specifically, if possible, a simple comparison of the gamma SWE values derived from total gamma counts used here and specific gamma radioelements (e.g., previous studies were used) would be valuable for potential readers.

*Peck, E. L., Bissell, V. C., Jones, E. B., & Burge, D. L. (1971). Evaluation of snow water equivalent by airborne measurement of passive terrestrial gamma radiation. Water Resources Research, 7(5), 1151-1159.*

You raise an important point about the decision to use total gamma count rates rather than specific radioisotope components. We provided a clearer justification for this approach inline with the previous

comment in the discussion section, incorporating a comparison of gamma SWE values derived from total gamma counts and specific gamma radioelements, as you suggested in the supplement. This comparison will help to contextualize our approach within the broader field of study.

3   Regarding the overestimation of SWE in Grassland in the second winter, I agree with the discussion point (L340-348) that a melt event earlier can contribute to the SWE uncertainty. To further discuss, I think it would be helpful if a time series of soil moisture and SWE throughout the season can be provided (if the station data is available near the sites). In this context, it might also be helpful to include a time series of temperature, snow depth, and precipitation at a GWFO station in the main body or as supplementary material for a better understanding of site weather conditions before and after the surveys.

We understand the need to elaborate on the overestimation of SWE in the second winter. We already provide some meteorological information on the most relevant midwinter melt event and to add visualisations of the time series we felt would add additional length for limited extra value to the reader. Unfortunately we did not observe the soil moisture conditions at this particular grassland site during the study period.

**Detailed comments**

L10-11 I suggest removing 'over shallow snowcovers' from the statement. Airborne applications using passive gamma-ray surveys have the capability to measure relatively moderate or deep snowpacks. The operational NOAA airborne gamma survey provides up to 1000 mm of SWE (https://www.nohrsc.noaa.gov/snowsurvey/historical.html; Mortimer et al., 2024).
*Mortimer, C., Mudryk, L., Cho, E., Derksen, C., Brady, M., & Vuyvich, C. (2024). Use of multiple reference data sources to cross validate gridded snow water equivalent products over North America. EGUsphere, 2024, 1-31.*

We revised this statement as suggested to reflect the broader capabilities of airborne applications using passive gamma-ray surveys.

L81-82 A recent effort to address the issue of the antecedent soil moisture change for the airborne program using SMAP soil moisture was presented in Cho et al. (2020). It would be good to mention it here.
*Cho, E., Jacobs, J. M., Schroeder, R., Tuttle, S. E., & Olheiser, C. (2020). Improvement of operational airborne gamma radiation snow water equivalent estimates using SMAP soil moisture. Remote Sensing of Environment, 240, 111668.*

Thank you for pointing out the study by Cho et al. (2020). We mention this work to provide context on addressing soil moisture changes here.

Section 2.1 It would be much more helpful if general snow conditions are provided including annual SWE (maximum and range), snow covered duration, and onset/offset dates with some literatures if available.

We provide more information on general snow conditions for this region with a reference to the literature.

L118 snow (water equivalent) → SWE

corrected

L141-142 For the stubble survey, how much did you expect that the larger GPS error ranges might impact the gamma SWE estimation? Please provide some justification regarding this.

We don't expect the larger GPS error for the stubble survey would have had a significant impact because this site was surveyed at a higher altitude (therefore larger footprint) – thus the GPS error would be partially compensated for by the coarser resolution. This is articulate in the edited manuscript in the discussion.

Figure 1. Have the authors considered providing some additional photos of both sites showing different surface characteristics along with the descriptions provided? I believe that would be informative for readers to better understand the sites.

Due to logistical limitations due to Covid protocols we do not have great additional photos of the sites/conditons to augment this unfortunately.

L201 Should it be the equation "2"?

Corrected

L205 high resolution → high spatial resolution

Corrected

Figure 3 How were the mean biases calculated? What is the reference count rate to calculate mean bias at a certain integration time? Please describe it briefly. Also, why were the shapes of changes in mean biases with change in integration time different particularly between Fall Stubble and Spring Stubble? Were the changes negligible in terms of the amount of count rate?

Upon reconsideration we felt that mean bias did not communicate the signal stability/noise that we initially had in mind. In contrast, the CV of the integration time (right panel) clearly expresses the SNR to integration time relationship much more clearly and we removed the mean bias (left panel).

Result 3.3 Please add the units of RMSE and mean bias (Figure 4) and the mean and uncertainty (Table 3). Also check the units throughout all figures and tables.

Units for RMSE, mean bias, and mean SWE values are added and checked throughout all figures and tables.

L426 Please include the areal mean SWE values with the uncertainty for each season.
L426 2.5-hectare → 2.5 x 106 m2
L432 22.5 m spatial resolution
L432 Include the mean SWE value along with RMSE

L118, L201, L205, L426, L432: These specific line comments were addressed accordingly.

Once again, we appreciate your constructive feedback and look forward to improving our manuscript with your suggestions.

Reviewer #2: Chris Donahue

We are pleased to hear that you find the manuscript well-written and suitable for publication in *The Cryosphere*, and we greatly appreciate your suggestions for improvement. We are committed to addressing each of your comments to enhance the quality and clarity of our work.

 General comments:
1       The main weakness of the study is the application of the manual density measurements in the reference dataset, the gamma and lidar fusion dataset, and in the defined uncertainty for the reference dataset. Figure 3 highlights large variation in density over the surveyed areas and simply taking the average density to represent the entire area introduces bias. Since this is the basis from which the SWE map is assessed, why were the locations of the density measurements considered? Were GPS points taken for each of the density measurements? I think a spatially interpolated density map would greatly improve the robustness of the analysis and the results for both the gamma SWE and gamma + lidar SWE. As for the uncertainty in the reference SWE, equation 4 describes the error in a single density measurement but does not represent the error propagation to SWE as a result of taking the average density over the survey area. The error in any given pixel should additionally contain the spread (e.g., 95% percentile) in density measurements over the survey area relative to the mean.

From previous work on snow density–depth relationships for prairie snow covers (shallow, windblown, and highly spatially variable) it has been demonstrated that, for snow depth < 60cm, density does not vary significantly with depth. Since the study areas are relatively small and subject to similar conditions (mainly blowing snow processes) there is little expected benefit to increasing the complexity of the density representation beyond applying a mean observed density (Shook and Gray 1994).  In contrast, the uncertainty in measurement of the density of shallow snow is such that an interpolated map will add significant noise that can be avoided with an average representation.  Not all density observations had GPS locations taken, largely due to fieldwork/logistical constraints under COVID protocols, so we do not have a consistent GPS positioning to employ.  Regarding the uncertainty in the reference SWE, we don't quite understand the comment.  The uncertainty in reference SWE (Eq3) does consider the uncertainty of the average density applied (as calculated in Equation 4).  We have not changed Equation 4 accordingly.

Shook, K.R.;Gray, D.M., Determining the Snow Water Equivalent of Shallow Prairie Snowcovers., Presented at Eastern Snow Conference, (1994), 1:14

2       One of the main goals of the paper is to present a new instrument for measuring gamma rays, however the detailed description of the data collected by the instrument is not shown nor described in detail. Per the manufacturers website, the MS-1000 collects data for multiple radionuclides across spectral channels. Please include full details of the data collected and how the total counts C were calculated based on the collected data. An example of the collected data (potentially as a sub-figure in Figure 1) illustrating

the collected data would be a nice addition that will help the reader understand the data collected with the instrument.

We recognize the importance of providing a comprehensive description of the data collected by the UAV-gamma spectrometer. We now include more information about the how the total counts C were calculated and added a figure to Appendix A to show the total count differences between snowcovered and bare ground.

Units are missing in many locations throughout figures and tables, that I have noted below.

We acknowledge the oversight in missing units in various parts of the manuscript. We have checked the figures and tables to have the appropriate units clearly stated.

**Line-by-Line Comments**
Line 18: In the abstract the "reference dataset" is mentioned without context to what it is. I recommend briefly describing it here for clarity.

We clarified the reference dataset in the abstract.

Line 22: I think this statement is somewhat misleading. The gamma+lidar fusion approach did not improve the results "substantially" and in fact the average site wide SWE measurement was slightly worse compared to the gamma SWE measurement alone. What did improve substantially is the spatial resolution of the SWE distribution.

We clarified that the improvement is in the greatly increased spatial representation of SWE.

Line 25: Resolution is better described in terms of fine and coarse. I recommend that fine/coarse replace high/low when describing resolution.

Terminology updated throughout.

Line 82: Missing the word "of"
"Approaches to correct for overwinter changes require independent estimates of soil moisture change.."

Corrected

Line 90: include the term "gamma" in UAV spectroscopy/spectrometry.

Corrected

Line 95: The terms spectroscopy and spectrometry are used interchangeably in manuscript. I recommend choosing one of these terms and use it consistently, which should probably be spectrometry because that is the term used in the title.

Terminology updated throughout.

Line 115: change was to were.

Corrected

Line 116: "to" is not needed here.

Corrected

Line 124: northwest and southeast are one word.

Corrected

Line 128: snowpack should be singular in this context.

Corrected

Line 152: Per major comment 1, please provide more information on how the spectral measurement was turned into count rates.

Updated. See also our response to the other reviewer on this topic.

Equation 4: This equation describes the uncertainty for the density measurements, but it does not describe the uncertainty that is propagated to your spatial SWE reference dataset by using the average of the density measurements across the entire survey error. The uncertainty for any given pixel in the reference SWE map should be defined somehow based on the spread of the density measurements in Figure 2 if you decide to keep the analysis based on the average density.

We didn't fully understand this comment as we are propagating the uncertainty of the average density and uncertainty of snow depth estimates from the UAV at each pixel prior to propagation to the overall average SWE observed for the respective study areas. We have not changed the manuscript accordingly.

Line 202-203: How was the lidar data interpolated? DIB or interpolation? Please add here.

Clarified in the manuscript. The UAV-lidar data is at a very high density (~100 pts/m$^2$) and we utilised LAStools approaches to convert the irregular point cloud to a 0.25 m gridded representation via a TIN surface fitting approach. Rescaling from the 0.25 m base resolution to other resolutions used the mean value of the new/larger grids.

Section 2.3.3: I don't understand why this analysis was not done spatially instead of using the average height and the average SWE from the two data products. This seems like a much more robust analysis and would make for a much stronger sensor fusion analysis.

There is spatial variability in the fusion. The spatial variability comes from the UAV-lidar snow depth (0.25 m resolution observations). Only the density is an areal average per the approach/justification explained in the second major comment. This is clarified.

Table 2: Add units to table. Also please add a statistic that describes the spread of the density measurements (i.e., 95 percentile, standard deviation, etc.)

Updated

Figure 2: Snow density needs units on the y-axis and in the caption, I am not familiar with any convention that uses (-) as a means to describe a density.

Updated to be kg m^-3

Line 229: Consider reminding the reader here what CV stands for. It is defined much earlier in text and I had to go back to remind myself.

Updated

Figure 3: What are the units of bias in this figure (y-axis). Could you provide some more context as to why some of these are positive and negative.

Updated the figure to remove the mean bias panel as per other reviewers similair comments.

Figure 4: y-axis units missing on bias, RMSE, and n. Also is the red fall stubble line missing in the number of observations plot? If it is underlying one of the other lines please make visible by increasing the line width of the underneath line or using dashed line.

We have created a dashed line now to show both colors.

Line 256: Biases should be described as positive/negative.

Updated

263: "negative bias" used here, good!

Thanks

Table 2: Table missing units.

Updated

Figure 7: Consider modifying the y-axis label so that it is clear that this doesn't refer to snow density. Also, would it make sense to add the gamma_lidar fusion distribution here?

The addition of the gamma_lidar fusion here is a great suggestion. Updated

Line 306: P should be rho.

Updated

Figure 8: Make clear which way the difference is done (ie., gam_lid – ref OR ref – gam_lid)

Updated

Line 344; ""…snowpack density changes…""

Updated

Line 402: "be" not needed

Updated

Your detailed and insightful review has provided us with clear directions for improvement. We are confident that these revisions strengthened the manuscript and to increase the value of its contribution to the field. Once again, we appreciate your valuable feedback .

Sincerely, Phillip Harder, Warren Helgason and John Pomeroy